# Species-Specific Allometric Equations for Predicting Belowground Root Biomass in Plantations: Case Study of Spotted Gums (*Corymbia citriodora* subspecies *variegata*) in Queensland

Trinh Huynh [1,2,*], Grahame Applegate [1], Tom Lewis [1,3], Anibal Nahuel A. Pachas [1,3], Mark A. Hunt [4], Mila Bristow [5] and David J. Lee [1]

[1] Forest Research Institute, University of the Sunshine Coast, Maroochydore, QLD 4556, Australia; gapples@usc.edu.au (G.A.); tom.lewis@daf.qld.gov.au (T.L.); nahuel.pachas@daf.qld.gov.au (A.N.A.P.); dlee@usc.edu.au (D.J.L.)

[2] Forest Science Institute of Central Highlands and South of Central, Da Lat City 670000, Vietnam

[3] Department of Agriculture and Fisheries, Queensland Government, 1 Cartwright Road, Gympie, QLD 4570, Australia

[4] University of Tasmania, Churchill Ave, Hobart, TAS 7001, Australia; m.hunt@utas.edu.au

[5] Plant Health Australia, Canberra, ACT 2600, Australia; mbristow@phau.com.au

* Correspondence: Trinh.Huynh@research.usc.edu.au

**Abstract:** Spotted gum (*Corymbia citriodora* spp. *variegata*; CCV) has been widely planted, has a wide natural distribution, and is the most important commercially harvested hardwood species in Queensland, Australia. It has a great capacity to sequester carbon, thus reducing the impact of $CO_2$ emissions on climate. Belowground root biomass (BGB) plays an important role as a carbon sink in terrestrial ecosystems. To explore the potential of biomass and carbon accumulation belowground, we developed and validated models for CCV plantations in Queensland. The roots of twenty-three individual trees (size range 11.8–42.0 cm diameter at breast height) from three sites were excavated to a 1-m depth and were weighed to obtain BGB. Weighted nonlinear regression models were most reliable for estimating BGB. To evaluate the candidate models, the data set was cross-validated with 70% of the data used for training and 30% of the data used for testing. The cross-validation process was repeated 23 times and the validation of the models were averaged over 23 iterations. The best model for predicting spotted gum BGB was based on a single parameter, with the diameter at breast height (D) as an independent variable. The best equation BGB = $0.02933 \times D^{2.5805}$ had an adjusted $R^2$ of 0.854 and a mean absolute percentage error of 0.090%. This equation was tested against published BGB equations; the findings from this are discussed. Our equation is recommended to allow improved estimates of BGB for this species.

**Keywords:** allometric equation; belowground root biomass; cross-validation; spotted gum plantations; weighted nonlinear models

## 1. Introduction

The contribution of forest ecosystems has been widely recognized for conserving and enhancing carbon sinks and reducing global warming [1–4]. Although plantation forests comprise 3% of the world's total forest area [5], they play an important role in climate change mitigation through their capacity to absorb and store carbon [6], particularly where plantation forests are established on previously cleared land. Accurate estimation of forest biomass is important, as it provides data on ecosystem productivity, nutrient flows, and their contribution to the global carbon cycle [3]. However, accurate estimates of biomass is limited for many sites and species due to the lack of specific allometric equations, the most common methods used for biomass estimations [7]. A robust regression for carbon

sequestration is therefore needed to provide plantation owners with confidence and allow trading of carbon credits.

Root systems contribute approximately half of the carbon being cycled annually in various forests and they account for 33% of global net primary production [8]. Belowground biomass (BGB) is defined as the biomass of living roots, including coarse roots (>2 mm diameter) and fine roots (≤2 mm diameter) [9]. Understanding the interactions and relative distribution of the belowground and aboveground biomass is a key requirement to understanding plant productivity [10–12]. Roots are considered a vital carbon sink in terrestrial ecosystems [13,14]. Root systems contribute 10–45% of the total tree biomass, depending on species [15,16], with biomass located mainly in root crowns and coarse roots [16]. Hence, if root biomass is underestimated at a site (e.g., through the use of inappropriate biomass estimating equations) in forestry carbon projects, the terrestrial carbon stocks can also be greatly underestimated [17].

BGB can be estimated using a range of methodologies, including destructive sampling, non-destructive sampling, inferential measurements, modelling, and ground penetrating radar [18]. While destructive sampling allows the most precise and accurate predictions of biomass, this approach is relatively rare in forest ecosystems due to challenges associated with sampling roots [10,17,19]. Destructive sampling involving harvesting of roots is time-consuming and costly. Sampling BGB of large trees is particularly challenging as large roots can penetrate deeply into the soil and be widespread, thus requiring a large area of excavation [12,13,17,19–23]. Consequently, in most cases, BGB is predicted based on aboveground biomass (AGB) and pre-existing allometric equations or root-to-shoot ratios (RS) [18,19]. However, these approaches have limitations. The RS in forests varies considerably, for example Snowdon et al. [18] reports RS of 0.25 to 0.70 for open-medium forest and woodlands, whereas Keith et al. [24] reports RS of 0.154 to 0.199 for *P. radiata*. Thus, the RS may not be appropriate for specific species and site conditions. As highlighted by Paul et al. [19], the BGB estimated using predictions based on AGB led to an increase in mean absolute prediction error of 13% at the individual stand level in comparison to the use of allometric equations. The obvious limitations here indicate that, where high levels of accuracy are required, allometric equations must be established and root biomass derived through destructive harvesting, rather than applying RS [25–27].

Accuracy of biomass estimation depends on the availability of reliable regression equations to utilize forest inventory data [28]. There are a number of allometric equations that have been developed for estimating BGB in Australian forests, mostly based on destructive sampling in native eucalypt forests and environmental plantings [10,19,29–31]. There are, however, no available species-specific equations to allow prediction of BGB of spotted gum (*Corymbia citriodora* subsp. *variegata*, CCV), despite this species being widely planted in South Africa, Brazil, and Israel [32] and having a widespread natural distribution along the eastern seaboard of Australia. CCV has been planted in Queensland and northern New South Wales and is the most important commercial native hardwood species for high-quality timber in Queensland [33,34] with great potential for commercial purposes. The species is more likely to thrive and adapt to the variable changes of climate compared to other eucalypts [35]. Research trials have shown that due to superior adaptability, CCV displays desirable characteristics relative to other *Eucalyptus* and *Corymbia* species, such as growth, wood properties, a relatively high tolerance to drought, and resistance to pests and diseases [34,35]. They also grow on nutrient-poor soils and in rainfall regimes that fluctuate from 600 to 2000 mm [36]. CCV is also a fast-growing tree on marginal areas on coastal sites in South Africa. This species indicated excellent potential for commercial forestry on the Zululand coastal plain [37]. Therefore, formulating a species specific BGB model will provide a simple way for plantation owners (e.g., farmers, investors, and government) to reap the dual benefits of timber production and carbon storage (trading carbon credits). In this paper we aimed to: (i) develop allometric equations for *Corymbia citriodora* subsp. *variegata* to estimate belowground biomass in spotted gum plantations, using independent variables including diameter at breast height and tree height and empirical data measured

by destructively sampled trees; (ii) determine the most appropriate equation to allow accurate estimates of BGB for this species; and (iii) test the application of these equations against an independent dataset.

## 2. Materials and Methods

### 2.1. Study Sites and Plantation Establishment

The study was conducted at three CCV research trials (451D, 451G and 13PHY) established by Queensland Government between 2000 and 2002 on contrasting soil types (Table 1) in southeast Queensland, Australia (Figure 1).

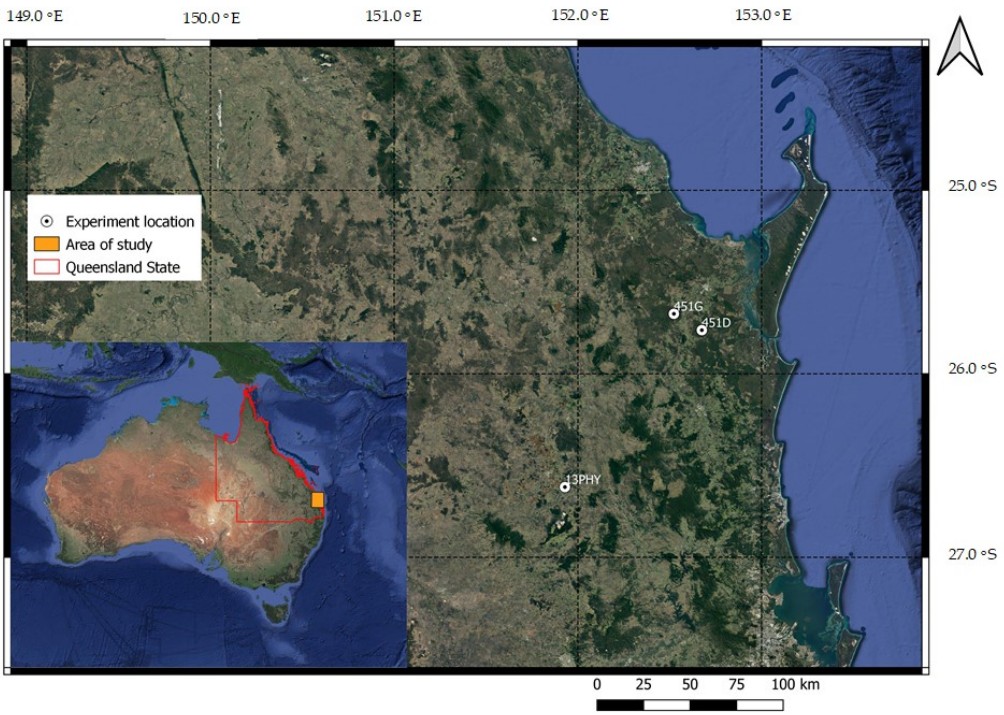

**Figure 1.** Locations of sampling sites in spotted gum plantations in Queensland, Australia. The site of 451D (1), 451G (2), and 13 PHY (3) are located in Tiaro, St Mary, and Coolabunia, respectively.

**Table 1.** Description of three plantation sites of *Corymbia citriodora* subsp. *variegate*.

| Description | 451D | 451G | 13PHY |
|---|---|---|---|
| Planting year | 2000 | 2002 | 2001 |
| Latitude | 25°45'45'' S | 25°40'24'' S | 26°37'4'' S |
| Longitude | 152°40'30'' E | 152°31'22'' E | 151°55'46'' E |
| Mean annual rainfall (mmyear$^{-1}$) | 1111 | 949 | 725 |
| Initial stocking (stemsha$^{-1}$) | 1000 | 1111 | 1000 |
| Current sampling (stemsha$^{-1}$) | 206 | 240 | 300 |
| Soil type [38] | Grey Kurosol | Red Ferrosol | Red Ferrosol and Brown Dermosol |
| Initial spacing (m) | 5 × 2 | 5 × 1.8 | 5 × 2 |

Site 451D is located in Bakers Logging Area, Tiaro State Forest, near Tiaro and was established in July 2000, covering an area of 10.75 ha. The soil type is a Grey Kurosol [38]; this acid, grey, texture-contrast soil is characterized by an apedal, lighter-textured A horizon overlying a structured heavy clay B horizon.

Site 451G is located in St Mary State Forest, covering an area of 5.5 ha. The soil type is a Red Ferrosol [38], which is an acid, red, well-structed soil without a clear texture contrast between A and B horizons.

Site 13 PHY is located within a 200 ha plantation established in May 2001, in the Burnett Valley at Coolabunia. This site presents two types of soil: Red Ferrosol (as per site 451G) and Brown Dermosol [38], which is an acid, brown, texture-contrast soil characterized by a weakly structed A horizon overlying as structured medium clay B horizon.

### 2.2. Sample Size

This study involved sampling belowground biomass of 23 trees (Table 2) from the most commonly planted provenance of the species (from the Gympie region). Sampling BGB across three sites aimed to capture the variability in tree sizes in terms of diameter at breast height (D) and total height (H). In 2009, 12 trees were sampled from experiments 451D, 451G, and 13PHY ranging from 7–9 years of age and from 11.8 to 18.2 cm D. In 2020, 11 trees were destructively sampled at sites 451D, with diameters ranging from 17.7–42.0 cm. The trees sampled in 2020 were divided into six diameter classes: (1) 15–20 cm; (2) 20.1–25 cm; (3) 25.1–30 cm; (4) 30.1–35 cm; (5) 35.1–40 cm; and (6) 40.1–45 cm to achieve an adequate spread of diameter classes in the sample [10,39,40]. In each diameter class, 2–3 individual trees were randomly chosen for sampling. Therefore, the total sample size comprised trees from across a wide range of D (11.8–42.0 cm), age classes (8–20 years old), and across three sites with different growth rates. An overview of the D and H distribution of the trees on these sites is given in Figure 2. The trees sampled were healthy and with single stems [41].

**Table 2.** Field sites, species information and descriptive statistics for predictor and response variables of 23 selected trees: D, diameter at breast height (1.3 m); H, tree height; RB, rootball; MR, medium roots; BGB, total belowground biomass; N, number of samples.

| Location (Age) | N | D (cm) | | H (m) | | RB (kg) | | MR (kg) | | BGB (kg) | |
|---|---|---|---|---|---|---|---|---|---|---|---|
| | | Min. | Max. | Min. | Max. | Min. | Max. | Min. | Max. | Min. | Max. |
| 451D (20) | 11 | 17.7 | 42.0 | 20.2 | 32.0 | 48.0 | 319.8 | 16.4 | 67.7 | 64.4 | 387.6 |
| 451D (9) | 3 | 12.0 | 17.8 | 16.5 | 17.5 | 13.1 | 49.6 | 2.8 | 5.8 | 16.0 | 55.4 |
| 451G (7) | 3 | 11.8 | 17.6 | 16.6 | 20.4 | 10.3 | 29.1 | 1.0 | 2.9 | 11.2 | 30.3 |
| 13 PHY (8) | 6 | 12.5 | 18.2 | 13.1 | 16.4 | 15.6 | 51.9 | 0.7 | 18.4 | 18.1 | 70.2 |
| Total | 23 | 11.8 | 42.0 | 13.1 | 32.0 | 10.3 | 319.8 | 0.7 | 67.7 | 11.2 | 387.6 |

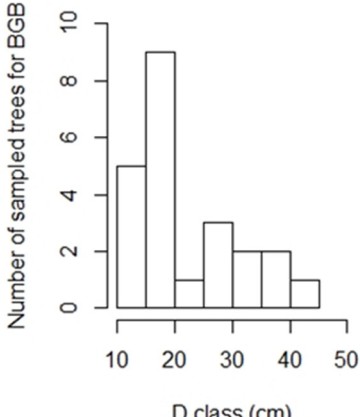
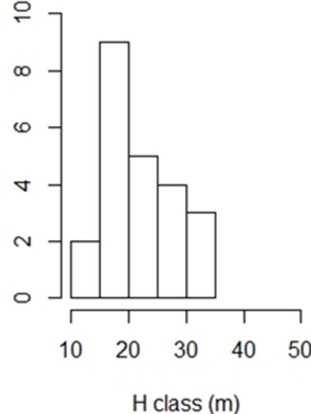

**Figure 2.** Distribution of diameter at breast height (D) and total height (H) of the 23 sampled trees used to develop the BGB allometric equations.

### 2.3. Destructive Sampling Procedure

A detailed methodology of the belowground biomass collection at the site 451D was reported by Huynh et al. [41]. A similar methodology was carried out at three sites (451D, 451G, and 13PHY) in the previous study in 2009.

In summary, each tree sampled was assigned a tree identification number (ID). The diameter over bark at breast height 1.3 m (D, cm) was recorded and the total height (H, m) was measured after the tree was felled.

A square plot of 1.5 × 1.5 m was marked around the tree stump with the tree in the center of the square. A second square measuring 2.5 m × 2.5 m was then marked out around the outside of the stump (Figure 3). In the 2009, a 1 m³ excavation area was used to sample the rootball monolith. However, in 2020, the trees were 11 years older with D up to 42 cm; hence, we extended the monolith sample area to ensure all of the large roots were included in the sample.

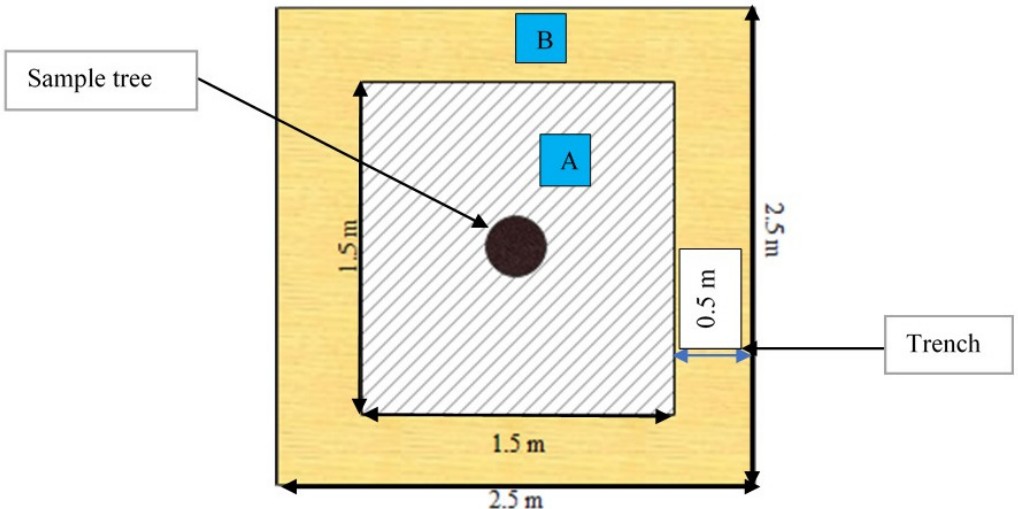

**Figure 3.** Excavated area: (**A**) The inner square (1.5 × 1.5 m) marked around sampled tree to sample rootball to a depth of 1 m, (**B**) the second square (2.5 × 2.5 m) located outside the rootball area to create four trenches (0.5 m wide around the inner square) and these trenches were dug to a 1-m depth to collect medium roots via soil sieving.

An 8-tonne excavator was used to dig four trenches down to 1 m deep prior to removing the rootball monolith. Soil in the excavated area was sieved through an 8 × 8 cm mesh with all roots removed (>2 mm). BGB in this study was defined as the rootball and medium roots (Huynh et al., 2020). The rootball included the stump to height of 50 cm above ground and to a depth of 1.0 m below the surface and enclosed by an area of 1.5 × 1.5 m. Medium roots (>2 mm diameter) were collected from soil immediately surrounding the rootball in four trenches, as shown in Figure 3. These roots were weighed and recorded as fresh BGB (kg).

After weighing the roots, sub-samples (at least 2 kg) of the rootball and medium roots were taken to the laboratory for dry weight determination. Root subsamples were oven-dried to a constant weight at 70 °C–100 °C. The fresh to oven-dry mass ratios were calculated and used to convert the total fresh mass of each root component into oven-dry mass.

*2.4. Data Analysis*

2.4.1. Biomass Model Development

Allometry involves linear or nonlinear modelling approaches to describe the correlation between biomass and tree variables (such as D and H) [42–44]. The power-law relationships are typical in many biomass equations [14,44–48]. Power models can be fitted using log-linear models or nonlinear models [44,47,49]. We applied both methods in our study. Firstly, we used power functions to analyze and develop allometric models with different independent variables (D and H). Secondly, we compared these equations based on a variety of selection criteria, including cross validation to select robust models and

to predict BGB based on forest inventory data. Finally, we selected the best models to accurately estimate BGB (as described in the section on model comparison and selection).

Within forests, most allometric equations have been developed using log-transformation (e.g., by Burrows et al. [50], Eamus et al. [8] and Paul et al. [19]) but if nonlinear models can account for heteroscedasticity there is no need to transform the parameter estimates [7,47]. The power-laws form $Y = \alpha X^\beta$, where $Y$, $\alpha$, $\beta$, and $X$ are the dependent variable, scaling coefficient, scaling exponent, and explanatory variable, respectively. If an additive error term or a multiplicative error term can be assumed for sampled trees, a simple BGB equation has the following form:

$$BGB = \alpha \times X^\beta \times \varepsilon \tag{1}$$

where BGB is total belowground biomass kgtree$^{-1}$; $X$ is tree variables D and H; $\alpha$ and $\beta$ are coefficients estimated by regression; and $\varepsilon$ is the error term [43,51]. However, as biomass data often exhibits a non-constant variance, Equation (1) was therefore transformed into a linear form using the natural logarithm (ln) of both sides of the equation prior to developing predictive models as shown in Equation (2):

$$\ln(BGB) = \ln(\alpha) + \beta \times \ln(X) + \varepsilon \tag{2}$$

To determine BGB, the log–log regression equations were transformed back from the logarithmic scale to the arithmetic scale. Based on the existing literature [19,48,52–54], an initial set of candidate models (Equations (3)–(7)) were selected:

$$\ln(BGB) = \ln(\alpha) + \beta \times \ln(D) \tag{3}$$

$$\ln(BGB) = \ln(\alpha) + \beta \times \ln(H) \tag{4}$$

$$\ln(BGB) = \ln(\alpha) + \beta \times \ln(DH) \tag{5}$$

$$\ln(BGB) = \ln(\alpha) + \beta \times \ln(D^2H) \tag{6}$$

$$\ln(BGB) = \ln(\alpha) + \beta \times \ln(DH^2) \tag{7}$$

where BGB = total belowground biomass (kg/tree), D = diameter at breast height (1.3 m) in (cm), H = tree height (m), and $\alpha$ and $\beta$ are parameter estimates of the model. The resulting estimates of biomass using the log–log transformation usually leads to a negative bias [30,55]. When back-transforming the estimation of biomass, a systematic bias result, such as a correction factor (CF), should be used to remove the bias [56]. Consequently, all logarithmic models (Equations (3)–(7)) include a CF. The CF was calculated as follows:

$$CF = \exp(RSE^2/2) \tag{8}$$

where RSE is residual standard error. If the value of RSE is high, it results in a higher CF and the model is less reliable. The models which provide the best relationships have a CF close to 1 [25,55,56]. In this study, the five candidate linear regression models were fitted using 'lm( )' function in the statistical software R.

We also evaluated an alternative approach using weighted nonlinear modelling to develop the models for BGB, as this approach is considered better at correcting for bias when transforming biomass estimates from logarithmic equations back to the arithmetic scale [42]. Weighted nonlinear models had the following general form:

$$BGB = \alpha \times X_{ij}^\beta + \varepsilon_{ij} \tag{9}$$

where BGB = total belowground biomass (kgtree$^{-1}$); $\alpha$ and $\beta$ are the parameter of the model; $X_{ij}$ is the covariate D (cm), H (m), or combination of D and H for *i*th sampled

tree; and $\varepsilon_{ij}$ is the random error related to the $i$th sampled tree. The variance function Var $(\varepsilon_{ij})$ [43,51] was defined as:

$$\mathrm{Var}\left(E_{ij}\right) = \widehat{\sigma}^2 \left(\nu_{ij}\right)^{2\delta} \tag{10}$$

where $E_{ij}$ is the absolute value; $\widehat{\sigma}^2$ is the estimated error sum of squares; $\nu_{ij}$ is the weighting variable ($DH^2$, $D^2H$, DH, H, and D in this study) associated with the $j$th tree; and $\delta$ is the variance function coefficient to be estimated. The value of $\delta$ was calculated using varPower constructor in the R package.

To be consistent with the previous models (Equations (3)–(7)), the following nonlinear models were developed specifically using the same combination of variables:

$$BGB = \alpha \times (D)^{\beta} \tag{11}$$

$$BGB = \alpha \times (H)^{\beta} \tag{12}$$

$$BGB = \alpha \times (DH)^{\beta} \tag{13}$$

$$BGB = \alpha \times (D^2H)^{\beta} \tag{14}$$

$$BGB = \alpha \times (DH^2)^{\beta} \tag{15}$$

The five candidate nonlinear models (Equations (11)–(15)) were fitted by the weighted maximum likelihood procedure [29,57] using 'nlme' package in R software, and the model diagnostics were checked using the ggplot 2 package [58].

### 2.4.2. Model Comparison and Selection

Accuracy and reliability should be considered in selection of the best-performing allometric equations [26,55,59,60]. Models with the same response (dependent) variables (i.e., Equations (3)–(7), fitted by transformed data or Equations (11)–(15) fitted by weighted nonlinear analysis) were compared based on five fit statistics: (i) Akaike information criterion (AIC); (ii) adjusted $R^2$ (adj. $R^2$); (iii) average bias, used to estimate model errors and for calculating the difference between the estimate and the true value [61]; (iv) root mean square error (RMSE) [44,61]; and (v) mean absolute percentage error (MAPE) [61]. To check for possible outliers and assess the goodness-of-fit of models, the diagnostic plots were also used to choose the best performing models [62]. The optimal model will have the lowest AIC, average bias, RMSE, MAPE, as well as a high adjusted $R^2$, low levels of collinearity [48], and will fit the data well.

$$AIC = -2\ln(L) + 2p \tag{16}$$

where L is the likelihood of the fitted model and p is total number of parameters in the model.

$$\mathrm{Bias} = \frac{1}{n}\sum_{i=1}^{n}(yi - \widehat{y}i) \tag{17}$$

$$\mathrm{RMSE} = \frac{1}{n}\sqrt{\sum_{i=1}^{n}(yi - \widehat{y}i)^2} \tag{18}$$

$$\mathrm{MAPE} = \frac{100}{n}\sum_{i=1}^{n}\frac{\left|yi - \widehat{y}i\right|}{yi} \tag{19}$$

where n is the number of samples; $yi$, $\widehat{y}i$, and $\overline{y}$ are observed, fitted/predicted and averaged value for the $i$th sample. Models with different dependent variables or weights were compared using Furnival's index (FI) [42,63,64]. The FI is calculated as follows:

$$\mathrm{FI} = \frac{1}{(f'(Y))}\sqrt{\mathrm{MSE}} \tag{20}$$

where $f'(Y)$ is the derivative of the dependent variable with respect to total BGB; MSE is the mean square error of the fitted equation, and the square bracket (( )) is the geometric mean. The five transformed models were compared with five weighted nonlinear models where more reliable models have lower FI values.

### 2.4.3. Model Cross Validation

To assess the applicability of the model for general use to predict BGB, a comparison between the two methods (linear and nonlinear) was therefore undertaken via cross-validation to select the best 'predicting model'. A Monte Carlo cross-validation procedure (MCCV) [65–67] was used to test the robustness and predictive value of each regression model. This procedure involves several steps; for each equation, the dataset was randomly split into two parts with 70% for training and 30% for testing [44,68]. The process was repeated 23 times with different random selections with the criteria for the selection of the models based on an average over 23 iterations. Comparison models were also based on the same criteria in Section 2.4, including AIC, adjusted $R^2$, percent bias, RMSE, and MAPE [61]. Finally, a model with the lowest errors was selected as follows:

$$\text{Bias} = \frac{1}{R} \sum_{r=1}^{R} \frac{100}{n} \sum_{i=1}^{n} \frac{(yi - \widehat{y})}{yi} \tag{21}$$

$$\text{RMSE} = \frac{1}{R} \sqrt{\sum_{r=1}^{R} \sum_{i=1}^{n} (yi - \widehat{y})^2} \tag{22}$$

$$\text{MAPE} = \frac{1}{R} \sum_{r=1}^{R} \frac{100}{n} \sum_{i=1}^{n} \frac{\left|yi - \widehat{yi}\right|}{yi} \tag{23}$$

where R = number of resampling (23); yi is measured BGB; and $\widehat{y}$ is estimated BGB from the cross-validation study.

## 3. Results

### 3.1. Descriptive Statistics

The D ranged from 11.8 cm to 42.0 cm across all sites and ages, while the H varied from 13.1 m to 32.0 m. The BGB of the sampled trees ranged from 11.2 to 387.6 kgtree$^{-1}$ (Table 2) and varied among tree ages. Root biomass increased with age. For example, BGB ranged from 11.2 to 70.2 kgtree$^{-1}$ in nine-year-old trees, whereases it ranged from 64.4 to 387.6 kgtree$^{-1}$ in the 20-year-old trees. The rootball accounted for approximately 80% of the BGB, and medium roots contributed about 20%.

Before fitting the allometric equations, data were checked for normality and a scatter plot was prepared to explore the relationships between BGB and independent variables D and H. The BGB–D relationship presented the best fit on the original scale (Figure S1).

### 3.2. Regression Equations Fitted to Natural Log Transformed Data

Separate transformed models for BGB developed based on D and H, or DH, $D^2H$, and $DH^2$ are presented in Table 3. Comparative plots between predicted and observed BGB are shown in the Supplementary Materials (Figure S2). The five candidate models had similar goodness of fit statistics among the different variables. The results showed that the predictor ln (D) (Equation (3)) alone performed poorest based on AIC and adj. $R^2$, whereas the combination $DH^2$ (Equation (7)) performed the best. A comparison of five criteria was used to distinguish the best model (AIC, adj. $R^2$, Bias, RMSE and MAPE), which showed that H (Equation (4)) was a poorer predictor than $D^2H$ (Equation (6)) and DH (Equation (5)). Analysis of AIC, adj. $R^2$, and RMSE for models with a single variable (D or H) showed that Equation (4) was superior to Equation (3). However, Equation (3) had lower values for bias and MAPE and a CF closer to 1 than Equation (4), and the spread of residuals from Equation (3) was narrower than that from Equation (4) (Figure S2). Values of CF for logarithmic models following back-transformation ranged from 1.035 to 1.096,

while the mean value was 1.055. Comparisons of the log transformed models are provided in Figure S2 and Table 3.

**Table 3.** Parameter estimates and their standard errors for BGB models developed based on logarithmic transformed models and weighted nonlinear models: D is diameter at breast height (cm), H is total tree height (m), CF is correction factor, AIC is Akaike's information criterion (kg), averaged bias (kg), averaged RMSE is averaged root mean square error (kg), averaged MAPE is averaged mean absolute percent error (%), δ is the variance function coefficient. FI = Furnival's Index. FI can be used to compare logarithmic transformed models and weighted nonlinear maximum likelihood models.

| Equation No. | Model Form | Parameter Estimates | | CF | Weight Variable | AIC | Adj. $R^2$ | Bias | RMSE | MAPE | FI |
|---|---|---|---|---|---|---|---|---|---|---|---|
| | | $\alpha$ | $\beta$ | | | | | | | | |
| | Logarithmic transformed models | | | | | | | | | | |
| (3) | $\ln(BGB) = \ln(\alpha) + \beta \times \ln(D)$ | 0.02354 | 2.64328 | 1.035 | - | 234.2 | 0.875 | −6.3787 | 37.738 | 22.179 | 16.306 |
| (4) | $\ln(BGB) = \ln(\alpha) + \beta \times \ln(H)$ | 0.00302 | 3.32052 | 1.096 | - | 227.9 | 0.905 | −1.9043 | 32.876 | 39.416 | 26.668 |
| (5) | $\ln(BGB) = \ln(\alpha) + \beta \times \ln(DH)$ | 0.00757 | 1.50922 | 1.047 | - | 222.5 | 0.928 | −4.2685 | 27.987 | 25.677 | 18.898 |
| (6) | $\ln(BGB) = \ln(\alpha) + \beta \times \ln(D^2H)$ | 0.01101 | 0.96482 | 1.040 | - | 226.9 | 0.913 | −5.0478 | 30.780 | 23.487 | 17.478 |
| (7) | $\ln(BGB) = \ln(\alpha) + \beta \times \ln(DH^2)$ | 0.00531 | 1.04515 | 1.058 | - | 220.5 | 0.934 | −3.4478 | 26.824 | 29.197 | 20.918 |
| | Weighted nonlinear models | | | | | | | | | | |
| (11) | $BGB = \alpha \times D^{\beta}$ | 0.02933 | 2.5805 | - | $1/D^{\delta}$ | 206.9 | 0.903 | −0.00003 | 0.014 | 0.029 | 0.014 |
| (12) | $BGB = \alpha \times H^{\beta}$ | 0.00269 | 3.3789 | - | $1/H^{\delta}$ | 226.8 | 0.906 | 0.00093 | 0.065 | 0.132 | 0.066 |
| (13) | $BGB = \alpha \times (DH)^{\beta}$ | 0.01150 | 1.44752 | - | $1/(DH)^{\delta}$ | 209.9 | 0.944 | −0.00048 | 0.058 | 0.120 | 0.058 |
| (14) | $BGB = \alpha \times (D^2H)^{\beta}$ | 0.01687 | 0.92222 | - | $1/(D^2H)^{\delta}$ | 207.5 | 0.939 | −0.00033 | 0.037 | 0.079 | 0.037 |
| (15) | $BGB = \alpha \times (DH^2)^{\beta}$ | 0.00722 | 1.01661 | - | $1/(DH^2)^{\delta}$ | 214.0 | 0.940 | −0.00016 | 0.077 | 0.158 | 0.777 |

### 3.3. Weighted Nonlinear Maximum Likelihood Models

Weighted nonlinear models (Equations (11)–(15)) are presented in Table 3. For these models, the adj. $R^2$ ranged from 0.903 (Equation (11)) to 0.944 (Equation (13)). Most regression models were closely related to the predictors, but Equation (12) with H alone had a weaker relationship based on AIC and averaged bias, while Equations (11) and (14) were more closely fitted, with the lowest AIC, bias, RMSE, and MAPE. Although the nonlinear maximum likelihood BGB model with D alone (Equation (11)) had a smaller adj. $R^2$ than the model with predictor $D^2H$ (Equation (14)), its weighted residuals graph indicates the narrowest variation and spread of points (Figure S3).

### 3.4. Model Comparison and Selection

Comparison of the values using Furnival's index (FI) showed that the overall performance of weighted nonlinear models was better than log-linear models (Table 3). Models $D^2H$ (Equation (14)) and D (Equation (11)) had the lowest FI values (0.025 and 0.014, respectively) of all the models developed. These results were supported by the standardized residual plots. All five log transformed models had two outliers in the larger predicted BGB size class (Figure S2). The first is at D of 40.0 cm with total fitted BGB of 387.6 kg, and the second is at D of 42.0 cm with total BGB of 341.8 kg. These were the largest diameter trees sampled. For the best overall model in terms of FI (Equation (11)) (Table 3), there was no evidence that inclusion of large trees underestimated BGB in stands of 7–20 years old. Hence, further analysis was restricted to the nonlinear models.

### 3.5. Model Cross Validation

The Monte Carlo cross-validation (MCCV) procedure was used to test the observed and predicted BGB of the five nonlinear models (Equations (11)–(15)). Results of MCCV procedure are presented in Table 4. The cross validation showed that the differences between criteria were non-significant, except for adj. $R^2$, and for the models with D, DH, and $D^2H$. However, the BGB model using H alone had relatively high errors (averaged bias, RMSE, and MAPE) and significant differences compared to the errors of other models tested. The $D^2H$ model had slightly higher AIC, bias, and RMSE than the model only using D. The validated and predicted values for the different models are also provided in Figure S4.

**Table 4.** Average predicted errors of five weighted nonlinear models from Monte Carlo cross validation to select equations for BGB. The cross-validation procedure was run 23 times, 70% data used for training, 30% data used for testing, and the statistics for comparison and validation of the models were averaged over 23 realizations. Bold: selected model based on cross-validation statistics and diagnostic plots (Figure S4).

| Equation No. | Model Form | AIC | Adj. $R^2$ | Bias | RMSE | MAPE |
|---|---|---|---|---|---|---|
| (11) | BGB = $\alpha \times$ D $^\beta$ | 146.4 | 0.854 | 0.040 | 0.063 | 0.090 |
| (12) | BGB = $\alpha \times$ H $^\beta$ | 160.9 | 0.916 | 0.422 | 2.631 | 1.213 |
| (13) | BGB = $\alpha \times$ (DH) $^\beta$ | 149.0 | 0.921 | 0.194 | 0.308 | 0.346 |
| (14) | BGB = $\alpha \times$ (D$^2$H) $^\beta$ | 147.6 | 0.910 | 0.136 | 0.175 | 0.224 |
| (15) | BGB = $\alpha \times$ (DH$^2$) $^\beta$ | 151.6 | 0.922 | 0.223 | 0.748 | 0.574 |

## 4. Discussion

The most reliable ways for determining of tree biomass and terrestrial carbon is to destructively sample and measure all trees across the whole plantation estate [46,60,69]. However, it is unrealistic to destructively sample all standing trees to estimate biomass due to the difficulty, time, and high cost of sampling [30] and negative environmental impacts, particularly associated with excavating root systems [69]. Thus, allometric relationships are considered an alternative approach and are widely applicable [30]. Developing reliable allometric models, based on a sample of destructively sampled trees, allows easy transfer from forest inventory data (e.g., D and H measures) to biomass estimates [28]. Once established, allometric equations allow estimation of forest carbon and $CO_2$ sequestration through simple, non-destructive measurements, such as diameter and tree height [40,70]. Allometric equations also play an important role in estimating and predicting the amount of carbon that will be stored in forests in the future [61].

### 4.1. Model Fitting and Cross Validation

Results suggest that both methods (logarithmic transformed models and weighted nonlinear models) closely described the relationships between BGB and predictors (D and H). Each set of the five candidate models had relatively high adj. $R^2$. Their values ranged from 0.875–0.934 and from 0.903–0.944 in Equations (3)–(7) and Equations (11)–(15), respectively (Table 3). However, based on statistics and diagnostic plots, weighted nonlinear regression models fitted with relatively low error levels in comparison with logarithmic transformed models. Furnival's index and the weighted residuals plots suggested that nonlinear models had higher reliability. This result contrasts with the findings of Moore [42] and Fordjour and Rahmad [64] who reported that the log-linear models performed better than nonlinear regression models in radiata pine plantations and liana species in tropical primary and secondary forests. The log transformation modelling approach has been used by many authors (e.g., Eamus et al. [8], Kuyah et al. [71], and Paul et al. [19]) when developing biomass equations. In contrast, fewer studies have used the nonlinear modelling approach to develop allometric equations [7,61,72–74].

Despite nonlinear models showing high reliability, we undertook cross-validation to identify the model that is best supported by our data (Table 4). Results of cross-validation statistics indicated that model BGB = $\alpha \times$ H $^\beta$ (Equation (12)) was less accurate than model BGB = $\alpha \times$ D $^\beta$ (Equation (11)), which had excellent potential for predicting BGB. A wide variety of statistical models have been developed for estimating tree biomass [49]. However, relatively few studies use cross-validation, despite this being an important step in the development of predictive models [75]. For example, our equation based on the H variable alone had a higher adj. $R^2$ compared to the equation with D, but the adj. $R^2$ alone is inadequate to select the best model. This implies that if the equation with H alone was not checked by cross-validation, this could result in large prediction errors. In our cross-validation analyses, the simple model based on D alone performed better than all other models tested, even though it had a slightly lower adj. $R^2$ (Figure S4). This was also

consistent with the selection of this model based on FI and other criteria (AIC, bias, RMSE and MAPE) discussed previously.

### 4.2. Predictors for BGB Models

Biomass estimating equations often include diameter or height as independent variables. The literature indicates that most studies used D (over 80% of studies) for establishing allometric equations [49]. Our results also showed that BGB was closely related to D. This is consistent with Eamus et al. [8], Kuyah et al. [76], and Paul et al. [19]. Our results are also consistent with Sileshi [48] who found that biomass equations with the addition of the variable H led to an increase in adj. $R^2$. For example, the equation BGB = $\alpha \times (DH)^\beta$ (Equation (13)) had a higher adj. $R^2$ (0.944) in comparison with the equation with D alone (0.903) (Equation (11)) but the overall errors (AIC, bias, RMSE, and MAPE) of the equations with both D and H were higher than that with D alone. The optimal model selected in this study was therefore the nonlinear involving only D. This also avoids the issue of collinearity, thus addressing concerns raised by Sileshi [48].

### 4.3. Biomass Model Comparisons

There have been a range of studies involving the development of regression equations for predicting BGB in eucalypt plantations (Table 5). We evaluated these equations using our data. The errors (bias, RMSE and MAPE) of predicted BGB using equations developed for *Eucalyptus* spp., in studies by authors such as Eamus et al. [8], Resh et al. [77], and Saint-André et al. [72], are not suitable for spotted gum plantations. Application of these equations indicated that BGB was underestimated by approximately 40–60% for the same tree diameters, which ranged from 25–50 cm in diameter. This suggests that these previous models may require improvement for application in spotted gum plantations, ideally with the inclusion of greater sampling effort for mature trees or larger trees. Previous studies by Eamus et al. [8], Resh et al. [77], and Saint-André et al. [72] were based on trees with diameters ranging from 3–25 cm.

**Table 5.** Comparison of average errors of BGB equation fitted by nonlinear maximum likelihood and other BGB equations worldwide with different eucalypt species. N = Native; P = Plantation.

| Reference | Forest Type | Site | Species (Diameter Range, cm) | Bias | RMSE | MAPE |
|---|---|---|---|---|---|---|
| This study | P | Australia | *Corymbia citriodora* subsp. *variegata* (11.8–42) | −0.003 | 0.014 | 0.029 |
| Paul et al. (2019) | N, P | Australia | Mixed *Eucalyptus* spp. (1.1–139) | 8.4 | 29.6 | 26.5 |
| Kuyah et al. (2012) | P | Kenya | *Eucalyptus* spp. (3–102) | 32.2 | 42.2 | 34.2 |
| Eamus et al. (2002) | N | Australia | *Eucalyptus* spp. (3–25) | 8.4 | 58.5 | 42.2 |
| Resh et al. (2003) | P | Australia | *E. globulus* and *E. nitens* (10–25) | 61.2 | 78.0 | 61.2 |
| Saint-André et al. (2005) | P | Congo | *E. alba* (3–25) | 66.9 | 85.7 | 67.3 |

In Figure 4, we present three equations that best predicted the spotted gum BGB. (1) The allometric equation developed by Kuyah et al. [76] where BGB (kgtree$^{-1}$) = 0.029D$^{2.432}$. This equation was established for *Eucalyptus* species (*Eucalyptus camaldulensis*, *Eucalyptus grandis*, and *Eucalyptus saligna)* in Western Kenya and constructed from the data collected by 72 destructively sampling trees with DBH from 3–102 cm. (2) The BGB model of Paul et al. [19] where BGB = exp(2.212 ln(D) − 2.682) × 1.096. Paul's equation was developed from single-stemmed trees, mostly eucalypts (77%) in hardwood plantations, native forests, and woodlands, but also included other high wood density trees and *Pinus pinaster*. The samples were collected from 810 trees with D ranging from 1.1 to 139 cm. (3) The optimal model selected in the current study based on Equation (11). The

comparisons indicated that allometric equations developed in our study predicted the highest yield of BGB among previous regression models. The models of Kuyah et al. [76] and Paul et al. [19] resulted in similar estimates to our model for small diameter trees (from 10–20 cm), but the predicted BGB values tended to be lower for larger diameter trees (Figure 4). A possible explanation for the lower prediction when applying Kuyah et al. [76] and Paul et al. [19] equations to the current data is the difference in characteristics of species. Although these studies included diameters ranging from small to large trees, a tree with a D of 40 cm had a predicted BGB of about 250 kg based on Paul's equation [19], while the spotted gum with a D of 30 cm also had a BGB of 250 kg. The prediction line of the Paul equation lies below the observed data and the prediction line of Equation (11). This difference may be due to the fact that we focused on a single species at three sites, whereas many species and ecoregions across Australia were used in the Paul study [19]. The Kuyah et al. [76] study was conducted in agricultural landscapes in Kenya where the average annual rainfall and soil characters are quite different to those in our study area, and again a larger number of species were considered.

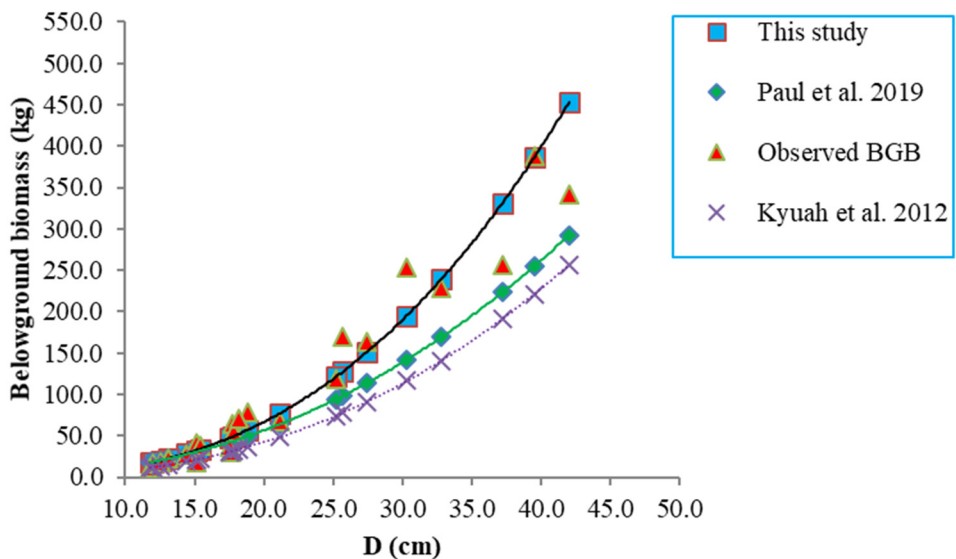

**Figure 4.** D versus total belowground biomass for 23 spotted gum trees from observed data and prediction lines applied for Equation (15) and equations of Kuyah et al. [76] and Paul et al. [19].

Several methods have been used for sampling BGB, such as manual excavation [10,72,78,79], soil pit excavation [80], monolith excavation [19,76,77,81], soil coring [77,80], water pressure [82], and compressed air [83,84]. In this study, we used an 8-tonne excavator to excavate biomass of individual trees to a 1.0-m depth [41]. The key benefit of this method is that root samples can be collected accurately, although fine roots (<2 mm in diameter) were not sampled during the excavation process. In prior excavations, Schenk and Jackson [85] reported that at least 50% of all roots were found in the upper 0.3 m of the soil profile, while 90% were located within 2 m of the ground surface. In our case, visual observations made during sampling suggested that excavation of an area of 2.5 × 2.5 × 1.0 m was adequate to sample most roots in a 20-year-old spotted gum plantation. The average errors of BGB equations discussed above implies that sampling methods may influence the resulting model parameters and their reliability. While our method may estimate BGB accurately, the main disadvantage of the approach is that it can be very destructive and labor-intensive. In addition, it is not easy to excavate large trees growing in soil with high bulk densities.

Due to time and funding constraints, our equations are based on 23 sample trees, across different ages, diameters, and three locations. It is recommended that future research expands on this dataset across different spotted gum plantations, with a focus on collecting more information on trees in the larger diameter classes. Inclusion of a greater number of

sites and sampling in native forest environments would allow the effect of environmental factors (e.g., soil characteristics) to be explored when developing allometric equations.

## 5. Conclusions

In this study, Equation (11), BGB = $\alpha \times D^{\beta}$, was the most suitable allometric equation for predicting belowground biomass of CCV plantations. This was confirmed by the Monte Carlo cross validation procedure. Use of this equation is advantageous as D is an easy and relatively accurate variable to measure in forest inventory assessments. Comparison of belowground biomass equations for various eucalypt forests suggests that our allometric equation differed from these equations and that use of earlier published equations may under-predict BGB for larger (>30 cm D) spotted gum trees. Therefore, development of the allometric equation in this study may contribute to more accurate predictions of belowground biomass and carbon stocks in spotted gum plantations in Queensland.

**Supplementary Materials:** The following are available online at https://www.mdpi.com/article/10.3390/f12091210/s1, Figure S1. Scatterplot matrix between belowground biomass (BGB) with diameter at breast height (D) and total height (H). Pearson correlation coefficient and significance (* = 0.05 to *** > 0.001) between the variables are shown above the diagonal; Figure S2. Relationship between belowground biomass (BGB, kgtree$^{-1}$) and tree size variables, including diameter at breast height, D (cm), and total height, H (m) based on natural log transformed data for the five candidate models (E3–E7 presents for Equations (3)–(7)). See Table 3 for criteria associated with these regressions. Above: predicted vs. observed BGB; and Below: standardized residual vs. predicted BGB; Figure S3. Allometric relationships between belowground biomass (BGB, kgtree$^{-1}$) and diameter at breast height, D (cm), and height above ground, H (m), based on models fitted by the weighted nonlinear maximum likelihood method. E11–E15 corresponds to five candidate equations (Equations (11)–(15)) shown in Table 3. Above: Observed vs. predicted BGB; and Below: Maximum likelihood weighted residuals vs. predicted BGB; Figure S4. Plots of five selected nonlinear models for belowground biomass (BGB, kgtree$^{-1}$) from the cross-validation with different predictor (s). E11–E15 validated to five candidate equations (Equations (11)–(15)) shown in Table 3. Validation data provided a random split of 70% for training and 30% for testing to predicted BGB. The dataset was repeated over 23 times to validate the performance of five nonlinear allometric equations developed and compared with the selected BGB models fit by maximum likelihood.

**Author Contributions:** T.H.: Conceptualization, Methodology, Formal analysis, Data curation and writing original draft. G.A.: Conceptualization, Investigation, Writing—review & editing, Supervision. T.L.: Conceptualization, Investigation, Writing—review & editing, Supervision. A.N.A.P.: Investigating, Writing—review & editing. M.A.H.: Investigating, Writing—review & editing. M.B.: Investigating, Writing—review & editing. D.J.L.: Conceptualization, Investigation, Funding acquisition, Writing—review & editing, Supervision. All authors have read and agreed to the published version of the manuscript.

**Funding:** This research was funded by the joint The Ministry of Education and Training, Vietnam (MoET) and University of the Sunshine Coast (USC) (MoET-VIED/USC scholarship); Australian Government Research Training Program Scholarship (RTP scholarship); and an internal grant provided by Forest Industries Research Centre-Forest Research Institute, USC. The financial assistance for excavating root system was provided by David Lee's UEDA and Department of Agriculture and Fisheries (DAF).

**Institutional Review Board Statement:** Not applicable.

**Informed Consent Statement:** Not applicable.

**Data Availability Statement:** Data available from the USC Research Bank https://doi.org/10.25907/00084.

**Acknowledgments:** We thank HQPlantations for giving us access to the site and allowing us to destructively sample trees and excavate roots to develop allometric equations. We acknowledge the DAF for providing material and equipment for the fieldwork. We thank DAF's staff, including Tracey Menzies, Tony Burridge and John Oostenbrink who contributed to field sampling and laboratory measurement. We would like to thank Tracey for helping us dry samples in the laboratory during

travel restrictions of Covid-19 from March to June, 2020. We also thank the reviewers for taking time to provide their useful suggestions and comments.

**Conflicts of Interest:** The authors declare no conflict of interest.

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
