# Peer review of "Species-Specific Allometric Equations for Predicting Belowground Root Biomass in Plantations: Case Study of Spotted Gums (Corymbia citriodora subspecies variegata) in Queensland"

_forests, doi:10.3390/f12091210_

Round 1

Reviewer 1 Report

Great job!

In spite of quite complex matter of this paper, I found reading it pretty straightforward and easy. Congratulations on sound methodology and mathematics. It is also worth noticing, that selected model is easy to implement for field foresters. 

My remarks are mostly of editorial kind, and should be easy to fix.

Best regards

Author Response

We thank the reviewer for your useful suggestions and comments and have taken them onboard as detailed below. Please see the attachment.

Reviewer 2 Report

Paper deals with currently very interesting research topics – estimation of  biomass for selected tree species in selected region. Although IPCC use and recommend specific default values for trees and forests in all regions in the world, detailed regional data are always required.  Specifically, young plantations of spotted gums in the age to 20 years from eastern Australia are presented. I consider construction of allometric  biomass models as very usable, particularly belowground biomass. Especially by proposals of new afforestation´s or plantations as very good potential to reduce of climate change effects on appropriate localities over the world.
Aim of work I consider as very eligible. Comparison of presented data with other sources confirm importance of regional models, or older research. Presented paper is by my opinion good prepared, authors made suitable data survey.
Paper is designed in five chapters. Abstract I consider as adequate. Introduction is short and clear. Used Material and Methods is suitable. Handicap is, that authors have used only 23 destructively sampled trees. Number of samples is very low, but from cost point of view and presented results, I consider this number as adequate. Authors have present relatively short chapter Results with approximately two page with  5 subchapter and only 2 tables. I recommend increase presented results, for example too with more figures or graphs from exhausting surveyed research material.   Chapter Conclusion I consider as adequate. Appropriate is too number of references (86).
I recommend carefully read whole text and repair all errors. Latin name mus be write with cursive (row 15, 80). Correct anomaly, maybe  caused by automatic cross conection (row 105, 134, 314, 372). Adjust please references in text according to Journal requirements (row 109, 134).
I believe, that authors can improve general quality of presented papers, but I think, that this work after minor modifications and repairs fulfil requirements of Journal Forests.   

Author Response

(The authors gave the same response as above.)

Reviewer 3 Report

Species-specific allometric equations for predicting below ground root biomass in plantations: case study of spotted gums (Corymbia citriodora subspecies variegata) in Queensland

By Trinh Huynh et al.

The authors developed and evaluated the allometric models for Corymbia citriodora subspecies variegata following the destructive sampling approach on the basis of size-mass allometric technique. The topic is definitely important because the allometric method is a cost-effective and reliable  approach to estimate root biomass at stand level- an urgent requirement to assess the belowground carbon stocks. Therefore, the manuscript fits perfectly with the scope of forests. Although having spatial (geographic region, vertical and horizontal spreading of roots etc.), temporal (stand age) and methodological (to include fine root) limitations, the assumption that growth rate of coarse root is similar to the woody biomass- has been found consistent for many species. Therefore the approach of the manuscript, the statistical techniques used for model selection, comparison and cross validation are mostly appropriate.  I have only one major concern with the manuscript, which I think can be handled with modification of results and discussions sections.

The authors used two age groups of trees 8 and 20 years, and mixed up together to develop the models. The original idea of allometry based on the fact that relative growth rates of two organs are determinant of their metabolic rates which obviously change over the life cycle of the trees, consequently, stand age alters the biomass allocation. I would suggest to develop age-specific allometric biomass equations for 11 trees of 20 years old and 12 trees of 8 years old separately. I think this is crucial as Corymbia citriodora is a fast-growing species and samplings were made at the respective ages. All statistical procedures will be applicable for both group.

The English level of the manuscript is good. 

Minor comments:

  1. Line 28-29: “Based on the errors of predicted models, previous published equations underestimated the BGB by 40 – 60% in trees with diameter ranging from 25 – 45 cm”. This comparison should not be included in abstract (but Ok in discussions section), this is not the objective of the study. Rather the authors can include a concluding remark on the basis of objective (iii) test the application of these equations against an independent dataset (Line 99).
  1. Line 297: To perform Monte Carlo cross-validation, data should be checked for normal distribution.
  2. Line 107- 126: The information is already mentioned in Table 1. Just add horizon characterization in Table 1 and delete Line 107-126 please.
  3. Line 120: The location of 13PHY is more than 100 km apart from other two, so please make sure that climatic factors are not vary enough to influence the growth of the CCV.
  4. Figure 1: Poor quality.
  5. Line 184-187: Crown diameters (CD) were measured but no use as independent variable (in data/equation. As CD data are not available for all trees, better delete this variable.
  6. Section 2.4. Biomass model development (2.4.1. Line 195-264), 2.4.3. model cross validation: Should be rewritten precisely, not necessary to explain with many references.  
  7. Line 106, 134, 314: ERROR Ref???
  8. Line 439: Ref 72 & 76: Comparison between data of allometric models at continental scale are not acceptable (i.e between Australia and Africa).

Author Response

(The authors gave the same response as above.)
